# Intervention Hypothesis for Training with Whole-Body Vibration to Improve Physical Fitness Levels: An Umbrella Review

**DOI:** 10.3390/jfmk9020100

**Published:** 2024-06-06

**Authors:** Luca Petrigna, Alessandra Amato, Martina Sortino, Bruno Trovato, Marta Zanghì, Federico Roggio, Giuseppe Musumeci

**Affiliations:** Department of Biomedical and Biotechnological Sciences, Section of Anatomy, Histology and Movement Science, School of Medicine, University of Catania, Via S. Sofia 97, 95123 Catania, Italy; luca.petrigna@unict.it (L.P.); alessandra.amato@unict.it (A.A.); martinasortino97@gmail.com (M.S.); brunotrovato94@gmail.com (B.T.); marta.zanghi@phd.unict.it (M.Z.); federico.roggio@unict.it (F.R.)

**Keywords:** SOP, protocol, vibrating platform, systematic review, WBV

## Abstract

Whole-body vibration (WBV) is a training modality, and it seems to be a safe and efficient exercise especially to improve different aspects of physical fitness in different populations. The protocols for WBV are still not standardized. The difficulty in comparing the data confuses the real efficacy of this instrument. Consequently, the objective of this umbrella review is to analyze the protocols previously adopted and eventually to propose a standard operating procedure for WBV training. Systematic review and meta-analysis of randomized controlled trials on WBV were searched on the electronic databases PubMed, Web of Science, and Scopus until 18 March 2024. A quality assessment of the studies included has been performed. A total of 20 studies were included in this umbrella review and frequency, magnitude, and amplitude intensity data were recorded. Detailed information about the protocols (static or dynamic exercises, barefoot or with shoes, intensity duration, weekly frequency, and vibration characteristics) was also collected. WBV presents widely different protocols. Consequently, a standard operating procedure has not been proposed for WBV training. A hypothesis of intervention was instead written in which parameters for frequency, amplitude, acceleration, and training mode were proposed.

## 1. Introduction

Whole-body vibration (WBV) appears to be a safe, less tiring, and less time-consuming exercise training methodology [1]. It seems to be an ideal form of training for athletes, to improve neuromuscular function, and for the general population, to reduce the effects of aging on musculoskeletal structures [2,3]. It seems to also have positive effects on cognition [4], making this training modality interesting and attractive. Whole-body vibration acts indirectly through vibrations that stimulate the sensory organs of the musculature and the central nervous system increasing muscle activity [5,6]. Mechanical vibration seems to deform the soft tissues, activates muscle spindles, and stimulates the neuromuscular system to produce a reflexed muscle activation [2]. This reaction intersects both the monosynaptic and the polysynaptic pathways probably because the primary endings of the muscle spindle could be more sensitive to vibrations than the secondary endings or the Golgi tendon organs [2]. The neurophysiological response corresponds to higher muscle activation during vibration training if compared to voluntary muscular activity, and this is confirmed by electromyography [2]. According to the literature, training with the WBV methodology could also increase muscle activity, force and mass, and power [5,7,8,9]. In older adults, this training is effective for counteracting the loss of muscle strength and maintaining muscle performance [10,11,12]. WBV seems also a valid intervention in people with compromised health [8] such as pelvic floor muscle disorders [13]. This training methodology appears to have limited or no benefit in improving muscle strength in young and fit people [11].

The oscillatory shear stress of the vibrations seems to also mechanically stimulate bone mass, activating the osteoblasts and reducing the activity of the osteoclasts [9,14]. Vibration could also provoke specific hormonal secretion, such as an increased level of testosterone, with a possible influence on calcium-handling mechanisms in skeletal muscle [2]. Studies have highlighted positive outcomes of this training on skeletal responses, especially among older adults [14,15], by reducing the risk of osteoporotic fracture (and by improving neuromuscular function) [16,17]. Unfortunately, studies have found that WBV has no beneficial effects on bone mass in post-menopausal or elderly women [18,19]. The outcomes of WBV on overall flexibility in older adults are inconclusive [12]. Results are on cardiovascular health are contradictory, sometimes with improvements [9], at other times with no changes [8].

Knowledge on WBV is still unsatisfactory with inconsistent results about its efficacy [1]. The effects of WBV could be influenced by the training protocol in terms of vibration characteristics (method of application, frequency, magnitude, and amplitude) and intervention (training type, intensity, and volume), especially over long time periods [20]. Furthermore, different reviews have highlighted that WBV protocols present inadequate rigor and different procedures, participants, intervention characteristics, and vibration platforms adopted, influencing the conclusions and the comparability of the studies [5,8,16]. Often there is heterogeneity in the studies making it difficult to standardize the protocol [17]. The current literature on the topic, despite the interesting results extrapolated, highlights how the findings are subject to possible errors due to publication bias and inherent imprecision [21]. A recent systematic review highlighted how WBV seems to improve neuromuscular activation and explosive power [22]. On the other side, the authors also concluded that further study protocols are necessary to provide a standardized target for amplitude, frequency, type of vibration, and method of application [22]. Generally, the literature suggests creating a well-structured and safe protocol and studying the effects of long-term vibration exercise [2,10,23].

In other fields, standard operating procedures are adopted [24]. A standard operating procedure is a step-by-step explanation of a specific intervention [24]; in this case, it could be a detailed explanation of the protocol in terms of the parameters of WBV training. Because there is, currently, a lack of standard operating procedures in this field, it would be interesting to adopt this model also for WBV training. Consequently, the objective of the present umbrella review was to analyze different reviews with specific questions [25] focusing on WBV protocols and, eventually, to create standard operating procedures for WBV training.

## 2. Materials and Methods

The umbrella review based the method following the Preferred Reporting Items for Systematic Reviews and Meta-Analyses (PRISMA) guidelines [26].

### 2.1. Search Strategy

The literature search was performed in PubMed (NLM), Web of Science (TS), and Scopus, until 18 March 2024. The Scielo database was searched on 23 May 2024. The following keyword terms were matched through the Boolean operators AND or OR:

Keywords 1: whole body vibration; vibration; WBV; balance training; vibrating platform; vibration plate;

Keywords 2: review; meta-analysis.

This is the string adopted in the databases:

(“whole body vibration” OR Vibration OR WBV OR “balance training” OR “vibrating platform” OR “vibration plate”) and (review or meta-analysis)

### 2.2. Eligibility Criteria

Eligibility criteria for the population, intervention, comparison, outcomes, and study design (PICO-S) were followed. The population was composed of healthy individuals without restrictions of age, gender, and physical activity background. Studies were excluded if mental (i.e., neurodegenerative disease, intellectual disability, psychosis, or obsessive compulsive disorders) or psychological (i.e., personality disorder, somatic symptom disorder, dementia, or delirium) disorders were detected due to the possible limitations of this population in following the protocol. The intervention had to be performed on a vibration platform in an upright position. Only studies related to sports sciences, health contests and promotion, and physical exercise were considered. Comparison was with a control group, both sedentary or active, and pre- versus post-intervention. Outcomes had no eligibility criteria.

Considering that the intensity of a WBV intervention [27,28] depends on the frequency, amplitude, magnitude of the acceleration due to gravity, and the amount of time spent on the platform (minutes per session, weekly frequency, and months of intervention); all those parameters were considered in the evaluation of the WBV protocols.

The study designs of the included studies were systematic reviews and meta-analyses of randomized controlled trials. Only manuscripts written in English were included, and no limitations were adopted for the country of publication.

### 2.3. Data Sources, Studies Sections, and Data Extraction

The manuscripts were manually searched. They were stored in EndNote X8 for an automatic duplicate selection. In a second screening phase, two investigators worked independently and selected the reviews against the eligibility criteria based on the titles, the abstract, and on the full text. If there was disagreement between the two investigators, a principal investigator took the final decision. A flow diagram that summarizes the selection process is reported.

The following information was extracted and inserted into the table, including information related to the first authors and year, review methodology, databases screened, number of reviews included, objective of the study, risk of bias assessment and score, conclusion of the study, population screened, duration of the training (weeks), WBV training characteristics (vibratory characteristics and load: frequency (number of repetitions of oscillatory cycles per second, Hertz: Hz)), amplitude (difference between the stationary point and the highest value reached by the oscillating plate, millimeters: mm), acceleration (determines the magnitude, grams: g, or meters/second: m/s: m/s^2^ or g), and duration (exposure time: minutes or seconds) [5]. The information extracted from the manuscripts was descriptively summarized.

### 2.4. Quality and Risk of Bias Assessment

The quality of the included systematic reviews was assessed by adopting Assessment of Multiple Systematic Reviews (AMSTAR) rating scale [29]. It is a reliable and valid instrument to assess the methodologic quality of systematic reviews [30]. It has 11 items that can be rated with 0 (no sufficient information available) or 1 (enough information). It characterizes the quality of systematic reviews at three levels, with 0–4 being considered poor quality; 5–7, moderate quality; and 8 and greater, high quality. All included reviews were scored independently by two authors and disagreements were resolved by the third author.

The risk of bias assessment will be performed with the Risk of Bias Assessment Tool for Systematic Review (ROBIS). It is a valid and reliable [31] instrument to assess systematic review. ROBIS is composed of three main sections, the first one is about the assessment of the relevance (facultative); the second one that wants to identify concerns with the review process; and the third one that wants to judge the risk of bias.

## 3. Results

A total of 36,209 studies (PubMed: 5148; Web of Science: 22,526; Scopus: 14,728; Scielo: 0) were initially included. After duplicate removal and title and abstract screening, a total of 282 studies were full-text screened. A final number of 19 studies were included in the umbrella review. The screening process is summarized in Figure 1.

Thirteen studies adopted PRISMA guidelines, three studies the Cochrane Handbook, and five studies had no information about this. The number of reviews included in the studies ranged from five to forty-six. The studies evaluated bone mineral density (No. of studies: six), muscle strength or strength-related characteristics (No. of studies: four), postural control (No. of studies: five), body mass and mobility (No. of studies: four), risk of falls (No. of studies: two), and gait capacity (No. of studies: two). The Physiotherapy Evidence Database (PEDro) scale was adopted in 10 studies while the Cochrane Handbook was adopted in 5 studies. Three studies had no information about the scale adopted for bias assessment. More details about the study characteristics are provided in Table 1.

### 3.1. Protocol Characteristics

The frequency ranged from 2 to 90 Hz. Most of the studies (No. of studies: six) adopted frequencies below 45 Hz and above (No. of studies: 12) 10 Hz. Frequencies above 50 Hz were adopted in seven studies. Studies on muscle mass adopted frequencies from 5 Hz to 60 Hz, most of the studies adopted minimal frequencies of 10 Hz, maximal frequency of 40 Hz (four studies), and 50 Hz (three studies). The oscillations mostly adopted ranged from 0 to 14 mm, with two studies that adopted a maximal oscillation of 8 mm. For the acceleration, this ranged from 0.05 to 32.2 g, with two studies that adopted values around 21 g. According to Chen and colleagues [33] the improvements are obtained despite the intervention parameters. The intervention protocol (F: 5–60 Hz; P: 0.05–14 mm; 0.05–32.2 g) of Lau and colleagues [39], and Rogan and colleagues [47] improved muscle strength. Muscle function and morphology were improved by Mikhael and colleagues [42] (F: 12–50 Hz; P: 0.2–8 mm; A: 0.1–22 g). Neuromuscular activation and explosive power improvements were detected by Reis-Silva and colleagues [45] (F: 10–40 Hz; P: 1.7–5 mm). No improvements in lean mass were detected by Rubio-Arias and colleagues [49] despite a similar intervention to the other studies (F: 12.5–40 Hz; P: 0–14 mm; A: 0.3–9.86 m/s^2^).

Studies that aimed to improve bone mineral density adopted frequencies from 12 Hz to 90 Hz, and two studies adopted 40 and 50 Hz. The oscillation ranged from 0.3 to 12 mm while the acceleration ranged from 0.1 to 22 g, three studies adopted 10 g as acceleration. According to the studies included, a frequency higher than 20 Hz induced a significant effect on the lumbar spine on bone mineral density [41] but also for frequencies from 12.5 Hz to 20 Hz [36], and from 12.6 Hz to 26 Hz [38]. Some studies obtained positive significant improvements in bone mineral density with higher vibration magnitudes of 3 g [36] and 8 g on bone mass at the lumbar spine [41]. Conversely, one review had positive significant outcomes with a low magnitude [40]. No significant results were obtained with high or low magnitude [50] or statistical differences were not between higher and lower amplitudes [41]. Platforms were mainly adopted that proposed vertical and side-alternating stimuli (sinusoidal) but also synchronous. Side-alternating vibrating platforms generate vertical left and right displacements, and they seem to have significant effects [36]. This kind of platform could add instability to train medio-lateral postural control [38]. In the synchronous platforms, the whole plate can oscillate up and down; no significant differences were observed [38]. No improvement was found for tilting platforms [36]. Participants stood in a static position (e.g., squat or lunge positions) [35], and this seems to work, especially in the bone mineral density lumbar spine and hip region [36]. In one review, most of the study participants stood in an upright position with slightly bent knees and feet shoulder-width apart [46].

Other studies obtained positive effects on functional mobility [32], postural balance and gait speed [35,46,47,48], and athletic performance [37]. Inconclusive results for postural balance and mobility were detected in two studies [38,44], and on the percentage of body fat [43].

One review suggested that training twice or thrice a week resulted in significantly better results [44,46,47,49], especially for bone mineral density [41]. A training session should consist of 3 to 10 series of 30–60 s WBV with a rest of 30 to 60 s in between [46]. The intervention can be continuous or with intermittent protocols [44]. The cumulative dose (time exposed to the vibration) was also found to be positively correlated to the effectiveness of WBV treatments [36].

Static positioning seems better to develop some aspects of muscle strength compared to dynamic exercises in older adults [34]. The type of exercise showed no significant differences between groups, but a static intervention produced positive effects on lower back bone mineral density [41]. All studies had static exercises; 14 studies also included dynamic exercises. The dynamic intervention consisted of lunge, squat, deep squat, wide stance squat, one-legged squat, lunge, toe stand, toe stand deep, moving heels, calf raises, left and right pivot in a front and lateral position, step up and down, toe stand, calves, knee extensor, and lateral weight transfer [42,46,47,48]. Studies proposed no differences between static and dynamic interventions [41,46].

The training on the platform could also be classified based on the population. Older adults were investigated by 10 studies. This population underwent training that ranged from 5 to 60 Hz. Different studies adopted a frequency of 40 Hz. The amplitude ranged between 0.5 and 14 mm. The magnitude ranged from 0.5 to 32.2 g. Intensity, amplitude, and frequency follow the trend previously described in the results section. Most of the studies (7 of 11) adopted dynamic exercises. A second population deeply investigated was postmenopausal women (four studies). Most of the studies on this population adopted frequencies that ranged from 12.5 to 40 Hz. One study arrived at 50 Hz. The amplitude ranged between 0 and 14 mm, while the magnitude was from 0.1 to 18 g. The remaining studies were on a mixed population. In this case, the frequency arrived at was 90 Hz, the amplitude was 20 mm, and the magnitude was 20 g.

WBV training was performed barefoot, wearing socks, or with shoes [44]. It was also suggested to use a handrail [44]. The results are summarized in Table 2.

### 3.2. Quality Assessment and Risk of Bias

The quality of the included studies has a mean of 7/11 with a range from four to ten. Within the included studies, the overall quality is mainly good/moderate but there are five studies with a Pedro score that is fair, one moderate-high, and one high. Four studies had no scores. The results of one review are unclear. One study has no results on the quality score. The results are summarized in Table 1 and Table 3.

The ROBIS tool detected that 11 studies presented a low risk of bias while 8 had an unclear risk of bias. Most of the studies provided insufficient eligibility criteria to minimize the risk of bias. Another point that presented a possible risk of bias is the methods used to identify and/or select studies, also in this case different studies presented an unclear risk of bias. More and much more detailed information is provided in Table 4.

## 4. Discussion

The protocols adopted in the literature are heterogeneous with wide differences in the WBV training and parameters. The frequency ranges were from low to high frequencies, and magnitude and amplitude also presented different intensities. Similarly, the authors adopted static and dynamic exercises, with different durations and weekly frequencies. This made it difficult to create a standard operating procedure, but we wanted to provide some suggestions to establish a common starting point. We created the indications on the basis of the data extrapolated from the included reviews. The frequency suggested for bone mineral density had to be below 20 Hz, the magnitude below 1 g, and the amplitude around 4 mm. According to one study, low-magnitude WBV can provide a significant improvement in bone mineral density [40]. According to another study, the magnitude should be greater than 3 g to overcome the damping effect produced by soft tissues and in order to effectively reach the target sites [36]. Even if a study suggests that a high frequency (>20 Hz) and lower amplitudes are effective in transferring the energy to the spine and hip [41], low-intensity magnitude (<1 g) and a low amplitude (<0.5 mm) seem to be safe, especially for individuals with a high risk of fractures [16,51]. Participants and trainers that have to execute this training have to know that frequencies below 20 Hz can cause chest pain, problems to the head and the jaw, and a feeling of discomfort [52]. For muscle strength, the frequency suggested had to be higher than 20 Hz, the magnitude below 1 g, and the amplitude around 12 mm. Higher frequencies and amplitudes enhance muscle power [53]. A recent review supports the fact that by increasing the frequency and amplitude, the beneficial effects are also increased, even if an ideal range has not been provided [54]. Also in this case, participants and trainers have to know that frequencies above 100 Hz could cause vascular and sensorineural dysfunctions and muscle stress and fatigue [55,56]. Our indications for muscle mass improvements are a proposal and it should be tested directly in the field; indeed, a study that adopted different frequencies, amplitudes, or magnitude detected no differences between the interventions [33]. A side-alternating vibration design is suggested because it seems to have significant effects [36]. This kind of vibration mode also has positive effects on neuromuscular activation [57,58], reducing the transmission of vibration to the head [59]. Training with the aim of reducing body fat mass and improving mobility or postural balance was similar. From the studies analyzed for postural balance and mobility [34,35,38,40,44,46,48], and for body composition [43,45,49], indications were proposed. More detailed information is in Table 4.

WBV training should be proposed in a static position, a squat or lunge position, feet shoulder-width apart, and barefoot. It has been decided to propose training in a static position because it seems that there are no differences between static and dynamic training [41,46]. A static WBV intervention is easier to propose and control. Furthermore, static intervention seems to produce positive effects, especially on lower backbone mineral density [41]. To standardize the position of the upper body, hands had to be positioned on the handrail. The posture suggested is with knees semi-flexed to limit the transmission of vibrations to the head [23]. Furthermore, despite full standing and hack squat seeming effective [36], semi-flexed knees present a significant difference in the bone mineral density of the lumbar spine, femoral neck, and trochanter and it is better than extended knees, which results in no significant differences [60]

For both interventions, it is suggested to train 2–3 times a week to obtain the best results [44,46,47,49], especially for bone mineral density [41]. Ideally, a training session should be composed of 3–10 series of 60 s and a comparable 60 s rest period in between [46]. It was decided to use bouts of 60 s because his seems to be associated with improvements in mobility or balance of institutionalized older people [32]. Ideally, to see improvements in muscle strength, at least 6 weeks are required, and this is comparable to other forms of active exercises (e.g., resistance training) [39]. The duration of sessions should be a longer length sessions (≥600 s) [41]. It should be taken into consideration that prolonged WBV training can have major negative effects on health, while shorter times seem to be safe [23]. Furthermore, if the vibratory stimulus is relatively short, it creates the potential for a more powerful and effective voluntary activation of skeletal muscle [2]. WBV should be implemented in an environment where supervision can be provided [14]. It is suggested to perform the test barefoot.

Because no important differences were detected in terms of frequency, magnitude, amplitude, number of series, and weekly training in the included studies (for more detailed information see the results section), the procedure proposed in Table 5 could be generalized for healthy individuals, regardless of age and sex.

The review presents several limitations. The first important limitation is related to the review typology. It is an umbrella review, a review of reviews, increasing the possibility of a wrong interpretation of the results. It would be better to analyze the literature with a systematic review or a scoping review, but the number of articles on this topic made this impossible. Furthermore, the manuscript protocol was not registered on PROSPERO as suggested by PRISMA, but the electronic database does not accept this article typology. The second important limitation of this umbrella review is the diversity of WBV protocols, which made it impossible to perform deeper analysis and interpretation of the results. One more limitation was related to the description of the same protocols, they were generally poorly described regarding the frequency, magnitude, amplitude, position on the platform, platform manufacture, footwear, and use of handrail for support. Another important limitation, despite trying to limit the risk of bias by including only systematic reviews that evaluated randomized controlled trials, was the poor quality of some works. Future studies should also focus attention on the proposal of the training for different age groups, also dividing the participants according to their sex and daily physical activity, sports practiced, or training routine selected.

Meta-analysis was also not performed due to the possible bias in the included reviews. Furthermore, some studies could be considered in two different reviews, influencing the results. Future studies should focus their attention on WBV in a microgravity environment and on disuse [23].

## 5. Conclusions

This review highlights that WBV training presents widely different protocols making it difficult to compare data. A standard operating procedure has not been proposed for WBV due to the differences in the protocols, but some specific indications are provided. The procedure suggested has been described and guidelines for frequency, magnitude, and amplitude have been proposed (Table 4). In this way, future studies can use the same methodology allowing them to compare different studies and clarify the ideas of WBV training efficacy.

## Figures and Tables

**Figure 1 jfmk-09-00100-f001:**
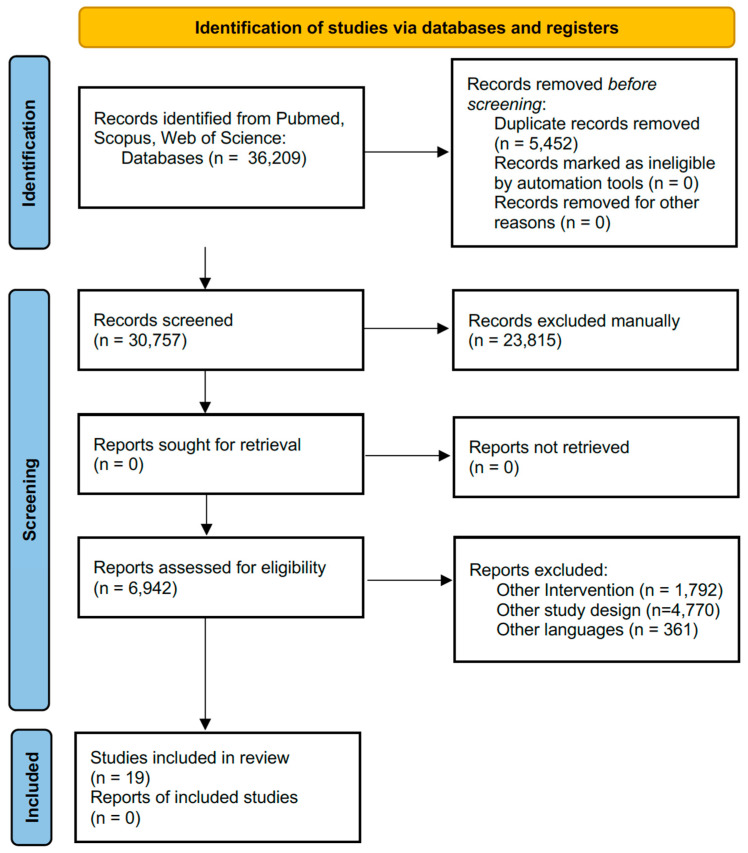
Flow diagram for the eligibility criteria process.

**Table 1 jfmk-09-00100-t001:** Characteristics of the included studies.

Author, Year	Guideline	Databases	N. of Reviews	Objective	Quality Score: Mean	Conclusion
Alvarez-Barbosa, 2020 [32]	PRISMA	AMED; CINAHL; Embase; Medline; PsycINFO; Scopus; Ebsco; WoK	10	Quantify the effect of WBV on balance, functional mobility, gait, functional performance, and quality of life	PEDro score: good	WBV could have benefits for functional mobility
Chen, 2017 [33]	PRISMA	WoK; Medline; Scopus; Embase; Cochrane Library	10	Evaluate the effects of WBVT on lean mass	Cochrane Handbook	WBV could improve lean or muscle mass in young adults. No dependency on the parameters, dose, and intervention
de Oliveira, 2023 [34]	PRISMA	Medline; Embase, CENTRAL, CINAHL, SPORTDiscus, WoK, LILACS, PEDro	35	Verify the effect of WBV on strength, power, and muscular endurance in older adults.	PEDro scale: fair	WBV increases lower-limb muscle strength but not power and muscle endurance
Fischer, 2019 [35]	PRISMA	Medline; Science Direct; Springer; Sage	46	Evaluate long-term effects of WBV training on gait	PEDro scale: fair	WBV training improves balance and gait speed in the elderly
Fratini, 2016 [36]	Cochrane Handbook, PRISMA	Medline; Cochrane Library; IEEE Xplore; Scopus; WoK	9	Evaluate the effect of WBV on bone mineral density	Possible bias	WBV treatments in elderly women can reduce BMD decline
Hortobágyi, 2015 [37]	NI	Medline, Web, WoK; and SportDiscus	21	Quantify the acute and chronic effects of WBV on athletic performance	PEDro score: fair	WBV has small and inconsistent acute and chronic effects on the athletic performance of athletes
Lam, 2012 [38]	NI	Medline; Excerpta Medica; CINAHL; Cochrane Library; PEDro; Science Citation Index	9	Effect of WBV on balance, mobility, and falls	PEDro score: fair	WBV on other balance/mobility outcomes and fall rates remains inconclusive. WBV seems effective in improving relatively balance ability and mobility, particularly in frailer subjects
Lau, 2011 [39]	NI	Medline; PEDro; CINAHL; Science Citation Index; Embase	18	Effect of WBV on bone mineral density and leg muscle strength	PEDro scale: fair	WBV is beneficial for enhancing leg muscle strength
Ma, 2016 [40]	Cochrane Handbook; PRISMA	Embase; WoK; Medline; Cochrane Library; China National Knowledge Infrastructure	8	Examine WBV effect on bone mineral density and fall prevention	12-item scale: moderate-high	Low-magnitude WBV therapy can provide a significant improvement in reducing bone loss in the lumbar spine. WBV can be used as an intervention for fall prevention
Marín-Cascales, 2018 [41]	PRISMA	Medline; WoK; Cochrane Library	10	Effect of WBV training on total, bone mineral density and identifying the potential moderating factors explaining the adaptations	PEDro scale: good	WBV is an effective method to improve lumbar spine BMD in postmenopausal and older women and to enhance femoral neck BMD in postmenopausal women younger than 65 years
Mikhael, 2010 [42]	NI	Medline; WoK; Scopus; SPORTDiscus; AMED AusportMed; CINAHL	6	Examine the effect of WBV on muscle or bone morphology and function	NI	Weak support for the efficacy of WBV exposure for muscle function, muscle morphology, or bone architecture
Omidvar, 2019 [43]	PRISMA	Medline; Embase; Cochrane Library; CINAHL	7	Describe the efficacy of WBV for reducing fat mass	Cochrane Handbook: unclear	Significant effect of WBV on total fat mass (kg); however, clinically insignificant effects on % of body fat
Orr, 2015 [44]	NI	NI	20	Effect of WBV on balance and functional mobility	PEDro score: fair to good	Some but inconclusive evidence for an overall effect of WBV on selected balance and mobility measures
Reis-Silva, 2023 [45]	PRISMA	Medline; Embase, WoK, Scopus	8	Effect of WBV on body composition in older adults	Cochrane Handbook: unclear	WBV seems to improve neuromuscular activation and explosive power but further research is required.
Rogan, 2011 [46]	PRISMA	Medline; PEDro; Cinahl; Cochrane Library	15	Summarize the current evidence for WBV interventions on postural control	Cochrane Handbook: moderate	Beneficial effect on dynamic balance
Rogan, 2015 [47]	PRISMA	Medline; Cochrane Library; PEDro; Other	38	WBV on strength, power, rate of force development, and functional strength	Cochrane Handbook: fair	Beneficial effects mainly in people not able to perform standard exercises
Rogan, 2017 [48]	PRISMA	Medline; Cochrane Library; PEDro; CINAHL	33	Effects of WBV on balance	Cochrane Handbook: fair	WBV can be used to improve static balance
Rubio-Arias, 2017 [49]	PRISMA	Medline; WoK; Cochrane Library	5	Valuate the effects of WBV on lean mass	PEDro scale: high	WBV alone may not be a sufficient stimulus to increase lean mass
Slatkovska, 2010 [50]	Cochrane Handbook	Medline; Embase; Cochrane Library; CINAHL; Ebsco; ProQuest	8	Examining effect of WBV on bone mineral density	Selection and attrition bias detected in most of the studies	Significant but small improvements in BMD in postmenopausal women and children and adolescents, but not in young adults

Allied and Complementary Medicine Database: AMED; bone mineral density: BMD; Cumulative Index to Nursing and Allied Health Literature; CINAHL; Excerpta Medica Database: Embase; Physiotherapy Evidence Database: PEDro; Psychological Information Database: PsycINFO; PubMed: Medline; SportDiscus: Ebsco; Whole-body vibration: WBVT; Web of Knowledge: WoK.

**Table 2 jfmk-09-00100-t002:** Characteristics of the intervention of the included reviews.

Author, Year	Population	Duration (Weeks)	Training	WBV Characteristics
Alvarez-Barbosa, 2020 [32]	institutionalized older adults	>6	2–3 sessions/week. Dynamic Vertical. 3–10 bouts of 30–60 s. Rest 60-s; Sinusoidal. 5 series of 15 s. Rest: 30 s	Vertical. F: 10–40 Hz; P: 3–7 mm; A: 1.6–2.2 g; Sinusoidal. F: 30 Hz; P: 2 mm
Chen, 2017 [33]	mix sample	>6	2–5 sessions/week of 4–20 min	F: 12.5–40 Hz; P: 0–14 mm; A: 0.3–12.9 g
de Oliveira, 2023 [34]	Older adults	>1	2–3 sessions/week of 2–30 min. Static	F: 5–60 Hz; P: 0.1–14 mm; 0.1–20.5 g
Fischer, 2019 [35]	mix sample	>4	2–5 sessions/week of 4–20 min. 1–135 sets per training session of 10–180 s. 3–300 s rest. Dynamic	F: 2–45 Hz; P: 0.4–20 mm
Fratini, 2016 [36]	postmenopausal women	NI	NI	F: 12.5–40 Hz; A: 0.3–18 g
Hortobágyi, 2015 [37]	Young	>4	2/3 session/week of 20–300 s	Vertical; side-alternating vibration. F: 25–45 Hz; P: 0.8–8.0 mm; A: 52–386 m/s^2^
Lam, 2012 [38]	older adults	>6	1–5 sessions/week from 1 to 27 sets of 15–180 s.	Vertical; side-alternating vibration. F: 10–54 Hz; P: 0.05–5 mm
Lau, 2011 [39]	older adults	>6	1–7 sessions/week Vibration usually delivered in bouts (1–27 of 30 s), with intermittent rest	F: 10–54 Hz; P: 0.05–8 mm; A: 0.05–32.2 g
Ma, 2016 [40]	postmenopausal women	NI	NI	F: 12.6–40 Hz
Marín-Cascales, 2018 [41]	postmenopausal women	>12	2–7 sessions/week of 90–1800 s Dynamic	F: 12.5–50 Hz; P: 1.5–12 mm; A: 0.2–20.12 m/s^2^
Mikhael, 2010 [42]	Older adults		1–7 sessions/week of 0.5–10 min. Continuous or intermittent. Dynamic	F: 12–50 Hz; P: 0.2–8 mm; A: 0.1–22 g
Omidvar, 2019 [43]	adult mix sample	>6	NI Dynamic	Vertical: F: 20–50 Hz; P: 2–6 mm Side-alternating: F 12–27 Hz; P: 1.5–4 mm
Orr, 2015 [44]	older adults	>6	Continuous (3–20 min); Intermittent. 2–100 min a week Dynamic	F: 6–40 Hz, A: 0.3–14.5 g; P: <0.1–8 mm
Reis-Silva, 2023 [45]	older adults	NI	Series of 15–60 s; 15–60 s rest Static-Dynamic	F: 10–40 Hz; P: 1.7–5 mm
Rogan, 2011 [46]	older adults	>6	3 sessions/week of 3–10 series of 30–60 s; 30–60 s rest Dynamic	F: 12–40 Hz; P: 0.5–8 mm
Rogan, 2015 [47]	older adults	>6	2–5 sessions/week of 1–12 sets of 15–90 s. 15–60 s rest Dynamic	Sinusoidal vertical F: 25–40 Hz, P: 2–4 mm. Sinusoidal side-alternating. F: 2.5–35 Hz; P: 0.05–12 mm
Rogan, 2017 [48]	older adults	>4	1–5 sessions/week of 2–15 sets of 15–72 s; 30–80 s rest. Dynamic	F: 12–40 Hz; P: 0.5–8 mm; A: 0.3 g
Rubio-Arias, 2017 [49]	postmenopausal women	>8	2–5 sessions/week of 300–1800 s	F: 12.5–40 Hz; P: 0–14 mm; A: 0.3–9.86 m/s^2^
Slatkovska, 2010 [50]	mix sample	>24	NI	F: 12–90 Hz

Frequency: F; amplitude: P; acceleration: A.

**Table 3 jfmk-09-00100-t003:** Quality assessment through the “assessment of multiple systematic reviews” (AMSTAR) if the included systematic reviews.

Study	1	2	3	4	5	6	7	8	9	10	11	Total
Alvarez-Barbosa, 2020 [32]	0	1	1	0	0	1	1	1	1	1	0	7
Chen, 2017 [33]	0	1	1	0	0	1	0	0	1	0	1	5
de Oliveira, 2023 [34]	1	1	1	0	0	1	1	1	1	0	1	8
Fischer, 2019 [35]	0	1	1	1	0	1	1	1	1	0	0	7
Fratini, 2016 [36]	0	1	1	1	0	1	0	0	1	0	0	5
Hortobágyi, 2015 [37]	0	1	1	0	0	1	1	1	1	1	0	7
Lau, 2011 [39]	0	1	1	0	0	1	1	1	1	0	1	7
Lam, 2012 [38]	0	1	1	1	0	1	1	1	1	1	1	9
Ma, 2016 [40]	0	1	1	0	0	1	1	1	1	0	1	7
Marín-Cascales, 2018 [41]	0	1	1	0	0	1	1	1	1	1	0	7
Mikhael, 2010 [42]	0	1	1	1	0	1	0	0	1	0	1	6
Omidvar, 2019 [43]	1	1	1	0	0	0	0	0	1	0	0	4
Orr, 2015 [44]	0	1	0	1	0	1	1	1	1	1	1	8
Reis-Silva, 2023 [45]	1	1	1	1	0	1	1	0	1	1	1	9
Rogan; 2011 [46]	0	1	1	1	0	1	1	1	1	1	1	9
Rogan; 2015 [47]	1	1	1	1	0	1	1	1	1	1	1	10
Rogan; 2017 [48]	1	1	1	1	0	1	1	1	1	1	1	10
Rubio-Arias, 2017 [49]	0	1	1	0	0	1	1	1	1	1	0	7
Slatkovska, 2010 [50]	0	1	1	0	0	1	1	1	1	0	0	6

Note: 1. Was an “a priori” design provided?; 2. Was there duplicate study selection and data extraction?; 3. Was a comprehensive literature search performed? At least two electronic sources, including years and databases used (e.g., Central, EMBASE, and MEDLINE); 4. Was the status of publication (i.e., the grey literature) used as an inclusion criterion?; 5. Was a list of studies (included and excluded) provided?; 6. Were the characteristics of the included studies provided?; 7. Was the scientific quality of the included studies assessed and documented?; 8. Was the scientific quality of the included studies used appropriately in formulating conclusions?; 9. Were the methods used to combine the findings of studies appropriate?; 10. Was the likelihood of publication bias assessed?; 11. Was potential conflicts of interest included?

**Table 4 jfmk-09-00100-t004:** Risk of bias assessment of the included studies.

	1	2	3	4	a	b	c	Risk of Bias
Alvarez-Barbosa, 2020 [32]	Unclear	Low	Low	Low	Low	Low	Low	Low
Chen, 2017 [33]	High	Unclear	Unclear	Low	Unclear	Low	Unclear	Unclear
de Oliveira, 2023 [34]	Low	Low	Low	Low	Low	Unclear	Low	Low
Fischer, 2019 [35]	Unclear	Low	Low	Low	Low	Low	Low	Low
Fratini, 2016 [36]	Unclear	Low	Unclear	Low	Unclear	Low	Unclear	Unclear
Hortobágyi, 2015 [37]	Unclear	High	Unclear	Low	Unclear	Unclear	Low	Unclear
Lam, 2012 [38]	Low	Low	Unclear	Low	Low	Low	Unclear	Low
Lau, 2011 [39]	Unclear	Low	Unclear	Low	Low	Low	Low	Low
Ma, 2016 [40]	Unclear	Low	Low	Low	Low	Low	Low	Low
Marín-Cascales, 2018 [41]	Unclear	High	Low	Low	Unclear	Low	Low	Unclear
Mikhael, 2010 [42]	Unclear	High	Unclear	Low	Unclear	Unclear	Low	Unclear
Omidvar, 2019 [43]	Low	Unclear	Unclear	Unclear	Unclear	Unclear	Low	Unclear
Orr, 2015 [44]	Unclear	Low	Low	Low	Low	Low	Low	Low
Reis-Silva, 2023 [45]	Low	Unclear	Low	Low	Low	Low	Low	Low
Rogan; 2011 [46]	Unclear	Low	Low	Low	Low	Low	Low	Low
Rogan; 2015 [47]	Unclear	Low	Low	Low	Low	Low	Low	Low
Rogan; 2017 [48]	Unclear	Low	Low	Low	Low	Low	Low	Low
Rubio-Arias, 2017 [49]	Unclear	Unclear	Low	Low	Unclear	Unclear	Low	Unclear
Slatkovska, 2010 [50]	Unclear	High	Low	Low	Unclear	Low	Low	Unclear

Note 1: Concerns regarding specification of study eligibility criteria; 2: Concerns regarding methods used to identify and/or select studies; 3: Concerns regarding methods used to collect data and appraise studies; 4: Concerns regarding the synthesis and findings; a: did the interpretation of findings address all the concerns identified in domains 1 to 4?; b: Was the relevance of the identified studies to the review’s research question appropriately considered?; c: Did the reviewers avoid emphasizing results on the basis of their statistical significance?

**Table 5 jfmk-09-00100-t005:** Guidelines for training on a whole-body vibration platform.

Goal	Frequency	Magnitude	Amplitude	Number of Series	Weekly Frequency
Improve muscle strength	20–60 Hz	>1 g	Around 12 mm	3–10 series of 60 s with 60 s rest	2–3 times
Improve bone mineral density	10–20 Hz	>1 g	Around 4 mm	3–10 series of 60 s with 60 s rest	2–3 times
Mobility and postural balance	10–40 Hz	>1 g	Around 8 mm	3–10 series of 60 s with 60 s rest	2–4 times
Body composition	10–50 Hz	0.3–9.86 m/s^2^	Around 6 mm	3–10 series of 60 s with 60 s rest	2–5 times
Generic indications	Adopt a static position, barefoot, feed shoulder-width apart, use a handrail. The vibration design suggested is side-alternating and sinusoidal. Supervision is required at the beginning				

## Data Availability

All data extracted are included in the tables and figures.

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
