# Peer review of "Intervention Hypothesis for Training with Whole-Body Vibration to Improve Physical Fitness Levels: An Umbrella Review"

_jfmk, 2024, doi:10.3390/jfmk9020100_

Round 1

Reviewer 1 Report

Comments and Suggestions for Authors

Dear Editor, thank you for the opportunity to review the article entitled “A Standard Operating Procedure proposal for a training with 2 the Whole-Body-Vibration to improve the physical fitness 3 level: an umbrella review”. The purpose of the work is interesting in investigating systematic reviews related to Whole-Body-vibration (WBV) and through these reviews defining the procedure proposal for a WBV training. The path adopted by the authors is consistent with the proposal of the article. However, some corrections are still necessary to the article.

1. Introduction

Authors often cite “studies” or “in the literature” for a particular answer or benefits of Whole-Body-vibration, however, they only cite one article. This requires referencing a greater number of studies.

Another point is to state some benefits based on the reference of just one article, the term “appears” or “investigated” would be more appropriate for this context.

Objective is clear and well described.

2. Materials and Methods

 The search strategy described in lines 77-78 “The literature search was performed in PubMed (NLM), Web of Science (TS), and 77 Scopus until 18 March 2024”. Perhaps searching in other data bases such as Scielo or Science Direct could find more systematic reviews on the topic. Even though the Systematic Reviews and Meta-Analyses (PRISMA) guidelines recommend that at least three databases be used as search strategies.

The rest of the Materials and Methods are very well described.

3. Results

I suggest in the figure to change and follow the Flow diagram for systematic review (PRISMA 2020). The Prisma 2020 flow diagram will be more complete.

I also suggest including risk of bias for the 20 studies included.

4.Discussion

The discussion is structured satisfactorily with the results found.

In line 315 to 313, the authors cite “WBV protocols are different and poorly 311 described about the frequency, magnitude, amplitude, position on the platform, platform 312 manufacture, footwear and use of handrail for support”. Because of this, the Figure 2 Standard Operating Procedure proposal for a WBV training. Couldn't it be compromised?

Within the systematic reviews analyzed, the authors were provided with a table summarizing the results found for each benefit of WBV training. Example: improved muscle strength (F: xx Hz; times a week, etc).

Author Response

Reviewer 1

Dear Editor, thank you for the opportunity to review the article entitled “A Standard Operating Procedure proposal for a training with 2 the Whole-Body-Vibration to improve the physical fitness 3 level: an umbrella review”. The purpose of the work is interesting in investigating systematic reviews related to Whole-Body-vibration (WBV) and through these reviews defining the procedure proposal for a WBV training. The path adopted by the authors is consistent with the proposal of the article. However, some corrections are still necessary to the article.

-Reply: Thank you for the comment. We would like to thank the Reviewer for the time spent on our manuscript and for the feedback and comments.

1. Introduction

Authors often cite “studies” or “in the literature” for a particular answer or benefits of Whole-Body-vibration, however, they only cite one article. This requires referencing a greater number of studies.

-Reply: Thank you for the comments. We corrected the sentences to avoid this problem.

Another point is to state some benefits based on the reference of just one article, the term “appears” or “investigated” would be more appropriate for this context.

-Reply: We corrected the introduction following this comment, thank you.

Objective is clear and well described.

-Reply: Thank you for the comment.

2. Materials and Methods

 The search strategy described in lines 77-78 “The literature search was performed in PubMed (NLM), Web of Science (TS), and 77 Scopus until 18 March 2024”. Perhaps searching in other data bases such as Scielo or Science Direct could find more systematic reviews on the topic. Even though the Systematic Reviews and Meta-Analyses (PRISMA) guidelines recommend that at least three databases be used as search strategies.

-Reply: we searched Scielo with our string and zero results were detected. Also adopting  “whole body vibration” did not bring interesting results, only 34 articles were detected, mainly in Spanish and original articles. Because we would like to be as systematic and objective as possible, in performing a random search with specific terms, we think it is not correct. About Science Direct, as Scopus, it is part of Elsevier. The difference between the two databases is the inclusion of more articles typology in Scopus. We think, also in this case, that to adopt three, the minimum number of articles suggested by PRISMA, but the most important databases in the sports science field, could provide us with a sufficient overview of the literature. We hope the Reviewer will accept our methodology.

The rest of the Materials and Methods are very well described.

-Reply: Thank you for the comment.

3. Results

I suggest in the figure to change and follow the Flow diagram for systematic review (PRISMA 2020). The Prisma 2020 flow diagram will be more complete.

-Reply: Thank you, we changed it.

I also suggest including risk of bias for the 20 studies included.

-Reply: we added the risk of bias assessment with the ROBIS tool. Thank you for the suggestion.

4.Discussion

The discussion is structured satisfactorily with the results found.

-Reply: Thank you for the comment.

In line 315 to 313, the authors cite “WBV protocols are different and poorly 311 described about the frequency, magnitude, amplitude, position on the platform, platform 312 manufacture, footwear and use of handrail for support”. Because of this, the Figure 2 Standard Operating Procedure proposal for a WBV training. Couldn't it be compromised?

-Reply: We agree with the Reviewer, the figure does not properly represent the findings. For this reason, we decided to remove it from the manuscript. Following this principle, we removed the definition of “standard operating procedure” from our proposal. We removed also it from the title. We instead proposed our guidelines as an intervention hypothesis. We hope the Reviewer will appreciate our decision.

Within the systematic reviews analyzed, the authors were provided with a table summarizing the results found for each benefit of WBV training. Example: improved muscle strength (F: xx Hz; times a week, etc).

-Reply: Following the reviewer's suggestion, we removed the figure 2 and added a more detailed table.

Reviewer 2 Report

Comments and Suggestions for Authors

Line: 2

The title

A Standard Operating Procedure proposal for a training with the Whole-Body-Vibration to improve the physical fitness level: an umbrella review

Comment 1

I am not sure that the paper can pass as an umbrella review. This is a review of reviews, but the topic is too broad, different: sample, protocols, research focuses... There are no PROSPERO registrations.

Line: 28

Introduction

Comment 2

The introduction talks too much about the effects of WBV training on the human body and too little about WBV training protocols, which are the main topic.

Line 89

The population was composed by healthy individual without restriction of age, gender and physical activity background.

DadeMatthews, 2022

Possible clinical benefit of WBV in improving existing osteoporosis treatment and reducing fracture risk

Oliveira, 2016

WBV is a potential coadjuvant in the prevention or treatment of osteoporosis, especially for BMD of the lumbar spine

Wang, 2022

WBV improve muscle strength and physical performance in older adults with sarcopenia

Wu, 2020

Vibration therapy could be a prospective strategy for improving muscle strength and physical performance in older adults with sarcopenia

Comment 3

These are not studies performed on healthy individuals. This needs to be better explained in the methods or studies done on subjects with a health issue should be removed.

Line 90-93

Studies were excluded if mental (i.e. neurodegenerative disease, intellectual disability, psychosis or obsessive-compulsive disorders) or psychological (i.e. personality disorder, somatic symptom disorder,  dementia or delirium) were mainly considered in the studies.

Comment 4

The studies on sample with mental disorders are excluded, studies on people with other diseases are not, why?

Line 143-146

The studies evaluated bone mineral density (n of studies: 8), muscle strength or strength related characteristics (n. of studies: 6), postural control (n. of studies: 5), body mass and mobility (n. of studies: 4), risk of falls (n. of studies: 2), and gait capacity (n. of studies: 2).

Comment 5

The question is which exercise protocol. Here you will find protocols of studies with a sample of all genders, age groups, physical condition, healthy and sick subjects (without mental disorders) in which the effects of exercise on many different aspects were investigated (mineral density, muscle strength or strength-related characteristics, postural control, body mass and mobility, fall risk of falls and gait capacity).

Given the diversity of the sample and the parameters observed, a scoping review would be more appropriate. Even the parameters related to the training protocol have numerous drawbacks. A scoping review is appropriate for such situations.

Line 226

3.2 Adverse effect

Comment 6

Adverse effects of exercise are not the subject of this review. They are not mentioned in either the introduction or the objectives. This is redundant.

Figure 1 - Other interventions, Other study design, Other languages, Other disciplines

Comment 7

In a chapter Eligibility criteria study design is partially explained, other language, interventions i disciplines are not mentioned. All the criteria by which you chose the studies must be explained in detail.

Line 97-101

Considering that the intensity of a WBV intervention [24,25] depends on frequency, amplitude, magnitude of the acceleration due to gravity, and amount of time spent on the platform (minutes per session, weekly frequency, and months of intervention); all those parameters were adopted in the evaluation of the WBV protocols.

Comment 8

It is not explained how these parameters are adapted.

Comment 9

Protocol is not registered with international prospective register of systematic reviews – PROSPERO.

Line 311-319

A limitation of this umbrella review is that WBV protocols are different and poorly described about the frequency, magnitude, amplitude, position on the platform, platform manufacture, footwear and use of handrail for support.

Comment 9

In such cases, a scoping review is more appropriate than an umbrella review

Line 321 – Conclusion

This review highlights that the WBV training presents widely different protocols making difficult to compare the data. A standard operating procedure has been proposed for WBV.

Coment 10

It is not clear to me how to recommend the same intensity or extent of a training regardless of gender and age or health condition. The same protocol is too easy for someone, while it may be impractical for another. Protocols should be standardized, but standardized protocols for different groups.

Author Response

We would like to thank the Reviewer for the time spent on our manuscript and all the corrections and comments

Line: 2

The title

A Standard Operating Procedure proposal for a training with the Whole-Body-Vibration to improve the physical fitness level: an umbrella review

Comment 1

I am not sure that the paper can pass as an umbrella review. This is a review of reviews, but the topic is too broad, different: sample, protocols, research focuses... There are no PROSPERO registrations.

-Reply: Thank you for the comment, unfortunately, PROSPERO doesn’t accept protocols for reviews of reviews. It has been impossible for us to register it. The term umbrella review wants to be a review of reviews but it is not mandatory to follow PRISMA statement. We wanted to follow it anyway to have a standardized methodology and improve the quality of the manuscript. We agree that there are differences in the sample, protocols… But we think that it is useful to highlight this problem. We hope the Reviewer understands the rationale behind our decisions.

Line: 28

Introduction

Comment 2

The introduction talks too much about the effects of WBV training on the human body and too little about WBV training protocols, which are the main topic.

-Reply: thank you for the comment, we added a part about the limits of the current literature in terms of protocol standardization.

Line 89

The population was composed by healthy individual without restriction of age, gender and physical activity background.

DadeMatthews, 2022

Possible clinical benefit of WBV in improving existing osteoporosis treatment and reducing fracture risk

Oliveira, 2016

WBV is a potential coadjuvant in the prevention or treatment of osteoporosis, especially for BMD of the lumbar spine

Wang, 2022 

WBV improve muscle strength and physical performance in older adults with sarcopenia

Wu, 2020

Vibration therapy could be a prospective strategy for improving muscle strength and physical performance in older adults with sarcopenia

Comment 3

These are not studies performed on healthy individuals. This needs to be better explained in the methods or studies done on subjects with a health issue should be removed.

-Reply: Thank you for this useful comment. It is correct, we did not consider people with pathologies. We included people with no physical, mental, or psychological disorders due to the limits that this population could get while using the vibration platform and the confounding factor that the pathology could be on the protocol. We removed the articles suggested by the Reviewer. We also checked for other articles included in our studies in which possible pathologies were in the population investigated. We hope the reviewer appreciates our decision the remove those papers with articles with people with pathologies..

Line 90-93

Studies were excluded if mental (i.e. neurodegenerative disease, intellectual disability, psychosis or obsessive-compulsive disorders) or psychological (i.e. personality disorder, somatic symptom disorder,  dementia or delirium) were mainly considered in the studies.

Comment 4

The studies on sample with mental disorders are excluded, studies on people with other diseases are not, why? 

-Reply: We removed the studies of people with mental and psychological disorders due to the possible limitations of this population in following the training routine. Same reason it has been taken for people with confirmed osteoporosis or sarcopenia. We now specified it in the text.

Line 143-146

The studies evaluated bone mineral density (n of studies: 8), muscle strength or strength related characteristics (n. of studies: 6), postural control (n. of studies: 5), body mass and mobility (n. of studies: 4), risk of falls (n. of studies: 2), and gait capacity (n. of studies: 2). 

Comment 5

The question is which exercise protocol. Here you will find protocols of studies with a sample of all genders, age groups, physical condition, healthy and sick subjects (without mental disorders) in which the effects of exercise on many different aspects were investigated (mineral density, muscle strength or strength-related characteristics, postural control, body mass and mobility, fall risk of falls and gait capacity).

Given the diversity of the sample and the parameters observed, a scoping review would be more appropriate. Even the parameters related to the training protocol have numerous drawbacks. A scoping review is appropriate for such situations.

-Reply: At the beginning, we wanted to perform a systematic review on this topic, but as the Reviewer has noted, the topic was wide. Consequently, we moved to a scoping review but we detected different systematic reviews on this topic, and to our knowledge, scoping review cannot include systematic reviews in the analysis. Consequently, because the systematic review read in the pre-screening was different fronting the topic from several points of view with different protocols and different conclusions, because the topic is still confusing, we wanted to provide a starting point, common and supported by the literature, to better understand the real efficacy of this instrument. To do so, we need, we suppose, a common and shared procedure, only in this way future works can be compared.

The Reviewer is right, there are different populations with different protocols, for this reason, because we believe that providing an overview of the topic could help the community to have a starting point, we decided to add a table creating different protocols for the improving-maintaining muscle strength, bone mineral density, body composition, and postural balance /mobility. Furthermore, we analyzed the training routine based on the population characteristics adding a section in the results and discussing it in the discussion. We hope the Reviewer will appreciate the work done to better valorize the findings.

Line 226

3.2 Adverse effect

Comment 6

Adverse effects of exercise are not the subject of this review. They are not mentioned in either the introduction or the objectives. This is redundant.

-Reply: Thank you for the suggestion, we removed it.

Figure 1 - Other interventions, Other study design, Other languages, Other disciplines

Comment 7

In a chapter Eligibility criteria study design is partially explained, other language, interventions i disciplines are not mentioned. All the criteria by which you chose the studies must be explained in detail.

-Reply: Thank you for the comment. We added information about the language. For the intervention, as we wrote in the method: The intervention had to be performed on a vibration platform in an upright position., only manuscripts that adopted the above characteristics were included. For the discipline, we added this information in the text: Only studies related to sports sciences, health contests and promotion, and physical exercise were considered. We hope now the section process results clearer.

Line 97-101

Considering that the intensity of a WBV intervention [24,25] depends on frequency, amplitude, magnitude of the acceleration due to gravity, and amount of time spent on the platform (minutes per session, weekly frequency, and months of intervention); all those parameters were adopted in the evaluation of the WBV protocols.

Comment 8

It is not explained how these parameters are adapted.

-Reply: I’m sorry, I think there is a misunderstanding. The parameters were not adapted, we considered them. Anyway, we changed the word “adopted” to “considered”.

Comment 9

Protocol is not registered with international prospective register of systematic reviews – PROSPERO. 

-Reply: PROSPERO only accepts systematic reviews and for us was not possible to register it. As we wrote in the method “The umbrella review based the method following the Preferred Reporting Items for Systematic Reviews and Meta-Analyses (PRISMA) guidelines” we wrote the protocol following PRISMA, but we did not write a systematic review.

Line 311-319

A limitation of this umbrella review is that WBV protocols are different and poorly described about the frequency, magnitude, amplitude, position on the platform, platform manufacture, footwear and use of handrail for support.

Comment 9

In such cases, a scoping review is more appropriate than an umbrella review

-Reply: As highlighted before, after a search, we found that it is suggested not to include reviews within a scoping review. This manuscript is only based on systematic review, so, to our knowledge, it is not possible to present it as a scoping review. If the Reviewer can support that the manuscript can be presented as a scoping review, we are more than welcome to change it in scoping review also adapting the protocol to PRISMA for scoping review.

Line 321 – Conclusion

This review highlights that the WBV training presents widely different protocols making difficult to compare the data. A standard operating procedure has been proposed for WBV.

Coment 10

It is not clear to me how to recommend the same intensity or extent of a training regardless of gender and age or health condition. The same protocol is too easy for someone, while it may be impractical for another. Protocols should be standardized, but standardized protocols for different groups.

-Reply: Thank you for this comment, we removed Figure 2, which is too generic, and we adopted a table in which specific indications according to the focus of the intervention.

Round 2

Reviewer 1 Report

Comments and Suggestions for Authors

Dear Editor, thank you for the opportunity to review the article again.

The authors responded satisfactorily to my suggestions and/or questions. Furthermore, they made the requested changes to the text.

Author Response

Thank you very much for the comments and help in the manuscript improvement.

Reviewer 2 Report

Comments and Suggestions for Authors The limitations of the study need to be expanded. Some responses I received from the authors point to limitations of the study.

Author Response

Thank you for the comment.

-The limitations of the study need to be expanded. Some responses I received from the authors point to limitations of the study.

Reply: We improved the manuscript limitations (please note the changing highlighted in yellow within the manuscript). We added all the limitations highlighted in the revision process. We hope the Reviewer will appreciate the changing.